# Fed MobiLLM: Efficient Federated LLM Fine-Tuning over Heterogeneous Mobile Devices via Server Assisted Side-Tuning

## Abstract

Collaboratively fine-tuning (FT) large language models (LLMs) over heterogeneous mobile devices fosters immense potential applications of personalized intelligence. However, such a vision faces critical system challenges. Conventional federated LLM FT approaches place prohibitive computational and memory burdens on mobile hardware, and their synchronous model aggregation protocols stall for slower devices. In this paper, we propose Fed MobiLLM, a novel design to facilitate efficient federated LLM FT across mobile devices with diverse computing/communication speeds and local model architectures. In particular, Fed MobiLLM implements a pioneering server-assisted federated side-tuning paradigm. Briefly, mobile devices perform lightweight forward propagation computations on local data using their frozen pre-scaled backbone LLMs, and then upload selected intermediate activations. The server trains a shared side-network independently, eliminating client-side backpropagation and enabling asynchronous updates. To bridge model heterogeneity across different devices, we introduce an adaptive layer-wise feature alignment method, which ensures consistent representations for collaboratively tuning a shared side network. Extensive experimental results demonstrate that Fed MobiLLM can maintain robust fine-tuning performance while achieving extremely low on-device memory, with at least 95.2% reduction in computation overhead, 93.2% reduction in communication costs and $5.1\times$ faster convergence compared to existing methods, validating its efficacy for practical LLM adaptation over heterogeneous mobile devices.

## 1 Introduction

Fine-tuning large language models (LLMs) for domain-specific tasks unlocks significant potential for novel applications, driving growing demand for personalized intelligence. Data required for such task adaptation is naturally generated and stored across massive personal mobile devices like smartphones and wearables. However, due to privacy constraints, this decentralized data cannot be combined for centralized training. Federated learning (FL) (McMahan et al., 2017) offers a promising paradigm for enabling collaborative, privacy-preserving LLM fine-tuning across mobile devices while keeping raw user data localized. While foundational models like GPT (Brown et al., 2020), BERT (Devlin et al., 2018), and LLaMA (Touvron et al., 2023) demonstrate broad capabilities (Ren et al., 2024; Ye, 2024; Brown et al., 2020), practical federated fine-tuning of LLMs faces critical bottlenecks due to mobile devices' limited computational power, memory capacity, and network bandwidth.

To tackle these resource constraints, recent work explores federated LLM FT with parameter-efficient fine-tuning (PEFT) methods like Adapters (Houlsby et al., 2019) or LoRA (Hu et al., 2022). These approaches follow the standard FL loop (local training → upload → server aggregation → model distribution), but exchanging only lightweight trainable modules (eg. LoRA) instead of full model weights. While reducing client-side computation (due to fewer trainable parameters), local training still requires storing LLM weights, intermediate activations and optimizer states—often exceeding the memory capacity of mobile devices. For example, tuning a 1.3B-parameter model with LoRA typically requires over 14.5 GB of GPU memory, which exceeds the 4–12 GB available on most mobile devices (Li et al., 2025). Furthermore, the inherently synchronous aggregation protocol forces the server to wait for multiple updates, resulting in significant resource waste when dealing with

heterogeneous devices; stragglers with slower computation or communication dramatically prolong convergence time—an issue amplified by the sheer size of modern LLMs.

In this paper, we propose Fed MobiLLM, a novel and efficient federated LLM fine-tuning framework built upon the server-assisted side-tuning principle inspired by PAE MobiLLM (Yang et al., 2025). Specifically, we decouple resource-intensive gradient computation from mobile devices by hosting all trainable parameters within a shared side-network on the server, while each mobile device retains only its frozen backbone LLM. During federated fine-tuning, each mobile device executes forward propagation computations on local data using its frozen backbone and uploads selected intermediate activations to the server. The server performs asynchronous backpropagation using these activations, computing gradients and updating the shared side-network independently for each mobile device's activations - without requiring global synchronization. In this way, Fed MobiLLM allows mobile devices to bypass costly on-device backpropagation and optimizer steps, drastically reducing client memory and computational load. Crucially, by enabling server-side side-network training to proceed immediately upon receiving any mobile device's activations, Fed MobiLLM eliminates the fundamental straggler bottleneck inherent in synchronous FL aggregation protocols. Fed MobiLLM thus offers a mobile device-friendly, training-efficient, and heterogeneity-tolerant solution for federated LLM adaptation across mobile devices.

Our salient contributions are summarized as follows:

- We propose Fed MobiLLM, a novel framework that pioneers mobile-friendly, asynchronous server-assisted side-tuning for LLM adaptation across distributed mobile data. Our design decouples computation by having devices perform only forward passes (no backpropagation) and upload activations, where the server asynchronously updates a unified side-network per-client activation arrival. This eliminates synchronization bottlenecks and removes all gradient computation from devices.

- We design adaptive mechanisms enabling Fed MobiLLM to support heterogeneous mobile devices via capacity-scaled backbone models and cross-architecture layer alignment techniques. This ensures devices with divergent model structures/sizes can collaboratively train a unified server-side shared side-network that consolidates knowledge from all devices.

- We implement and evaluate Fed MobiLLM across diverse mobile platforms (NVIDIA Jetson TX2, Xavier NX, and AGX Xavier) and model scales (sub-billion to billion parameters). Experiments across multiple tasks and system settings demonstrate that Fed MobiLLM achieves extremely low on-device memory usage, with at least 95.2% reduction in computation overhead, 93.2% reduction in communication costs and $5.1\times$ faster convergence compared to existing methods. It also delivers state-of-the-art and highly robust LLM fine-tuning performance.

## 2 RELATED WORK

### 2.1 FEDERATED LLM FINE-TUNING

The limited scale and diversity of data on individual mobile devices necessitate collaborative LLM fine-tuning across devices to enhance model performance. FL has emerged as a dominant paradigm for this purpose, where devices perform local training and upload parameter updates to a central server for aggregation. However, fine-tuning large language models (LLMs) under this paradigm presents significant challenges due to their substantial computational and memory requirements. To address these constraints, PEFT methods have been widely adopted for local training (Zhang et al., 2023). Techniques such as Adapters (Houlsby et al., 2019), LoRA (Hu et al., 2022), and BitFit (Zaken et al., 2021) freeze pretrained backbone parameters while fine-tuning only minimal additional parameters. To further alleviate on-device memory burden, split federated learning (SFL) approaches further reduce device load by offloading deeper layers to the server (Tian et al., 2022; Chen et al., 2025; Gupta & Raskar, 2018). Specifically, devices sequentially exchange activations/gradients with the server during forward/backward passes, while the device-side sub-models require periodic weight aggregations across devices. Alternatively, forward-only methods like FwdLLM (Xu et al., 2024) eliminate backpropagation by estimating gradients through parameter perturbations. While reducing activation memory to inference levels, this approach requires multiple forward passes per update - increasing device computational overhead.

## 2.2 Efficient On-device LLM Fine-Tuning

Enabling LLM fine-tuning directly on mobile devices requires innovative architectures that minimize memory and computational load while retaining raw data locally. Server-assisted side-tuning, pioneered by MobiLLM (Li et al., 2025), addresses this by decoupling trainable side-networks from frozen backbones and offloading all the gradient computation to the server. Advancing this approach, PAE MobiLLM (Yang et al., 2025)(as illustrated in Fig. 1) introduces key optimizations: mobile devices perform only a single forward pass through frozen backbones,

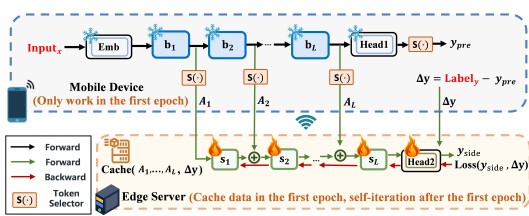

Figure 1: An overview of the PAE MobiLLM.

compute output deviations $\Delta y = \text{Label}_y - y_{\text{pre}}$ without exposing ground-truth labels, and upload selected sparse intermediate activations $(\mathbf{A}_1, \ldots, \mathbf{A}_L)$ alongside $\Delta y$. The server then trains the side-network exclusively using these activation-deviation pairs $(\mathbf{A}_1, \cdots, \mathbf{A}_L, \Delta\mathbf{y})$, ensuring no raw data access. A server-cached replay mechanism further reduces device overhead by limiting local data processing to the first epoch, with subsequent iterations handled server-side. This achieves an efficient balance between device resource consumption, communication costs, and training speed for on-device LLM fine-tuning. While highly effective for single-device scenarios, scaling server-assisted side-tuning to federated environments introduces fundamental new challenges, including coordination across heterogeneous devices and cross-architecture knowledge aggregation. Our work addresses this gap by extending the side-tuning paradigm to federated fine-tuning scenarios through novel architectural and algorithmic innovations.

## 3 Motivation

### 3.1 Inefficiencies of SOTA Federated LLM FT

State-of-the-art federated LLM fine-tuning methods fail to adequately address the tension between mobile device constraints and LLM computational demands. PEFT techniques reduce communication costs by updating only small modules (e.g., adapters, low-rank matrices), yet still require devices to perform backpropagation through full LLMs. This necessitates storing intermediate activations for all layers during fine-tuning, resulting in memory footprints that significantly exceed typical mobile device capacities - often causing out-of-memory failures or impractical computation delays. Split federated learning mitigates device load by offloading deeper layers to the server but introduces coordination bottlenecks: devices must serially exchange activations and gradients with the server during forward/backward passes, while device-side sub-models require periodic cross-device weight aggregation. In addition, in scenarios where the backbone model is privately deployed within a closed device cluster and is not publicly available, sharing its parameters with the server may violate confidentiality requirements. Forward-only perturbation methods (e.g., FwdLLM) avoid backpropagation at the cost of increased computational overhead, requiring multiple forward passes per update to estimate gradients. Collectively, existing federated LLM fine-tuning approaches exhibit critical gaps and cannot simultaneously optimize on-device memory usage, computational overhead, communication cost, and fine-tuning performance.

### 3.2 Heterogeneity Challenges for Federated LLM FT

The significant variation in computational power, memory capacity, and network bandwidth across mobile devices introduces fundamental limitations to traditional synchronous federated learning protocols, which require the central server to wait for model updates from all participating devices before every-round global aggregation. This synchronization barrier creates unavoidable delays caused by slow devices (stragglers), forcing faster devices to remain idle during waiting periods. When applied to LLM fine-tuning, this synchronization bottleneck is exacerbated: intensive computational demands further amplify performance gaps between high- and low-end devices, leaving powerful ones underutilized or idle for extended periods and significantly slowing overall progress. Moreover, memory heterogeneity also leads to significant resource waste. To enable cross-device parameter

aggregation, all devices must adopt the same backbone model, forcing its size to conform to the memory constraints of the least-capable device (Su et al., 2024). This constraint underutilizes the capacity of high-resource devices and results in computational waste. In practice, capable devices naturally prefer to load larger, more powerful models for achieving better performance. In summary, as device diversity increases, this inefficiency fundamentally conflicts with FL's core goal of collaborative resource utilization, and necessitates asynchronous paradigm designs resilient to device heterogeneity in computation and memory.

## 4    FED MOBiLLM DESIGN

### 4.1    FED MOBiLLM OVERVIEW AND PROCEDURE

Fed MobiLLM is a server-assisted distributed learning framework designed to enable resource-constrained mobile devices to collaboratively fine-tune LLMs using their local data. Fig. 2(a) presents an overview of the Fed MobiLLM system. The key idea is to let devices retain merely a frozen LLM backbone with pre-trained parameters locally while deploying a tunable side network on the server. This distinguishes it fundamentally from conventional federated FT approaches that require full LLMs (frozen backbone + tunable modules) on each device. Fed MobiLLM coordinates devices to extract features from local data via forward propagation through their frozen backbones, guiding the server-side training of a shared side-network. By centralizing all tunable parameters on the server, Fed MobiLLM eliminates expensive on-device backpropagation and reduces memory overhead (activations/optimizer states) during LLM fine-tuning.

During federated tuning, each mobile device performs forward propagation through its frozen backbone using mini-batches sampled from its local dataset. For each mini-batch, device $i$ transmits selected intermediate activations $(\mathbf{A}_i^1, \ldots, \mathbf{A}_i^L)$ from the backbone layers, along with the prediction residual $\Delta y_i$ (defined as the difference between ground-truth label and backbone output) to the server, consistent with PAE MobiLLM's design (Yang et al., 2025). Upon receiving these activation-deviation pairs $(\mathbf{A}_i^1, \cdots, \mathbf{A}_i^L, \Delta \mathbf{y}_i)$ from any device, the server updates the shared side-network immediately, i.e., processing each device's contribution sequentially upon arrival without global synchronization[1]. As device-side local models are frozen, Fed MobiLLM inherently yields the following advantages: i) Devices can perform local computations and upload activations at the same time. ii) Devices keep processing their local data without stopping to wait for server-side updates. iii) The server triggers immediate side-network updates upon receiving any device's activations, eliminating global synchronization barriers. iv) Each device processes its local dataset in just a single pass during the entire training process. Particularly, the server caches received activations to construct an activation repository for iterative side-network training. To mitigate non-IID data bias, cached samples are randomly shuffled during storage. During idle periods, the server trains continuously on cached samples to maximize computational efficiency. This design ensures uninterrupted training despite slow devices—eliminating straggler bottlenecks while maintaining full utilization of server resources. After all devices complete uploading, the server performs efficient standalone tuning using the comprehensive cached dataset. Through flexible and non-blocking device-server parallel collaboration, Fed MobiLLM eliminates training bottlenecks and progress stalls caused by slower devices. (A more detailed description of Fed MobiLLM's training procedure under heterogeneous devices is provided in Appendix B. )

### 4.2    HETEROGENEITY-AWARE CROSS-MODEL ALIGNMENT

Beyond computational heterogeneity through non-blocking device-server collaboration, Fed MobiLLM fundamentally resolves memory-driven model capacity divergence across mobile devices where deployable backbone sizes are dictated by each device's memory constraints. Specifically, Fed MobiLLM allows each device to load a pre-trained backbone model scaled to its hardware capacity.

---

[1]Note that the concerns about data privacy arising from the transmission of intermediate activations in Fed MobiLLM are aligned with the definitions adopted in Google's federated learning (FL) work (McMahan et al., 2017) and split learning related architectures (Tian et al., 2022). Similar to these approaches, Fed MobiLLM is naturally compatible with existing advanced privacy-preserving techniques, such as differential privacy (DP) mechanisms (Dwork, 2006) (e.g., adding DP noise to gradients, inputs, outputs, or objective functions) and secure multiparty computation (Du & Atallah, 2001).

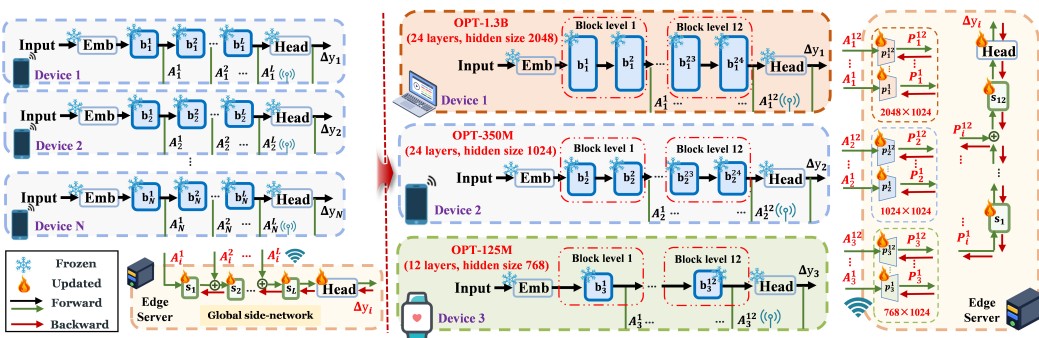

**(a) Fed MobiLLM in the Homogeneous Model Scenario**  **(b) Fed MobiLLM in the Heterogeneous Model Scenario**

Figure 2: An overview of the Fed MobiLLM.

As illustrated in Fig. 2(b), devices may employ different-sized models (e.g., OPT-125M, OPT-350M, OPT-1.3B (Zhang et al., 2022)), resulting in backbone architectures with varying numbers of transformer layers and hidden dimensions. This architectural diversity introduces two key challenges: (1) server-side adapters must accommodate diverse backbone structures, and (2) uploaded activations exhibit inconsistent shapes across devices. To resolve these issues, we introduce two structural alignment mechanisms—layer-wise activation sampling and hidden dimension scaling—that unify activation patterns across heterogeneous devices for federated training of the shared side-network on the server.

**Layer-Wise Activation Sampling.** Our layer-wise activation sampling mechanism addresses mismatches in transformer layer counts across backbone models. Inspired by empirical evidence in LST (Sung et al., 2022) demonstrating that not all layers require dedicated adaptation modules, we propose selectively extracting activations from strategic layer positions.

As shown in Fig. 2(b), when devices use backbones with differing layer counts (e.g., 12 vs. 24 layers), we partition all models into a fixed number of blocks (e.g., 12 blocks). Activations are then sampled exclusively from the final layer of each block. The server-side network is configured with an equal number of adapter modules, each processing one aligned activation block. Through comparative experiments of various strategies, we established optimal configuration guidelines: the block count is set to the minimum layer depth among participating models (e.g., 12 blocks for 12/24-layer models), while deeper models are partitioned at uniform intervals (e.g., sampling every other layer in a 24-layer backbone). This approach ensures layer-wise structural consistency while preserving representational capacity.

**Hidden Size Scaling.** Pre-trained models of different scales exhibit varying hidden sizes, preventing direct integration of their activations into a unified side-network adapter module. To resolve this, we introduce dedicated trainable linear projection layers for each backbone model type, mapping activations to a consistent hidden size for shared side-network processing.

As shown in Fig. 2(b), three distinct backbone models (hidden sizes: 2048, 1024, 768) require dimension standardization. We configure the side-network adapter with a target hidden size of 1024 and deploy projection layers $(p_i^1, \ldots, p_i^{12})$ on the server for each model variant. Consider device 1 using OPT-1.3B (hidden size 2048): its activations $(\mathbf{A}_1^1, \ldots, \mathbf{A}_1^{12})$ pass through corresponding projection layers $(\mathbf{p}_1^1, \ldots, \mathbf{p}_1^{12}) \in \mathbb{R}^{2048*1024}$, producing transformed activations $(\mathbf{P}_1^1, \ldots, \mathbf{P}_1^{12})$ with uniform 1024-dimensional features. These standardized activations then feed into the shared adapter modules $(s_i^1, \ldots, s_i^{12})$, which enable collaborative training across heterogeneous models.

All projection layers are server-managed and co-trained with the side-network. After fine-tuning, devices download their specific projection layers alongside the shared side-network for local inference. Our empirical study suggests selecting mid-range dimensions (e.g., 1024 for 768/1024/2048 scenarios) can optimize efficiency-accuracy balance in multi-device scenarios, while prioritizing larger dimensions can preserve representation capacity in scenarios involving only two model sizes.

## 5 EXPERIMENTAL SETUP

### 5.1 FED MOBILLM IMPLEMENTATION

The experimental testbed consists of a server with an NVIDIA A100 GPU and three types of heterogeneous client devices representing increasing computational capabilities for LLM processing: (1) NVIDIA Jetson TX2 (8GB RAM, 1.3 TFLOPS), (2) NVIDIA Jetson Xavier NX (8GB RAM, 6 TFLOPS peak), and (3) NVIDIA Jetson AGX Xavier (16GB RAM, up to 10 TFLOPS peak).

### 5.2 MODELS, DATASETS AND PARAMETERS

**Models:** To systematically evaluate Fed MobiLLM's performance, we employ two representative pre-trained LLM architectures: i) decoder-based OPT series (OPT-1.3B, OPT-350M, OPT-125M), and ii) encoder-based RoBERTa series (RoBERTa-large(350M) and RoBERTa-base(125M)). This selection ensures architectural diversity while maintaining mobile compatibility. All models are initialized via HuggingFace Transformers (Wolf et al., 2019).

**Datasets:** We take the GLUE benchmark (Wang et al., 2018) and DialogSum dataset (Chen et al., 2021) for the evaluation of NLP tasks, which are widely used in the fine-tuning research for LLM (Zhang et al., 2023; Sun et al., 2024; Sung et al., 2022). GLUE benchmark comprises eight tasks, including linguistic acceptability (CoLA (Warstadt, 2019)), sentiment analysis (SST-2 (Socher et al., 2013)), similarity and paraphrase (MRPC (Dolan & Brockett, 2005), QQP (Iyer et al., 2017), STS-B (Cer et al., 2017)), and natural language inference (MNLI (Williams et al., 2017), QNLI (Rajpurkar, 2016), RTE (Bentivogli et al., 2009)). DialogSum includes summaries of real-world conversations on a diverse set of topics and scenarios to evaluate text-generation tasks. We use ROUGE scores (R1/R2/RL) as the accuracy metric. Following FedPETuning (Zhang et al., 2023), we simulate non-IID data partitions using a Dirichlet distribution with concentration parameter $\alpha$, where lower $\alpha$ values induce higher label distribution shift. (See Appendix C for dataset details.)

**Parameters:** Following FedPETuning (Zhang et al., 2023), we set the number of communication rounds to 100 and the number of local training epochs to 1 for all baselines under the FL paradigm. All configurations deploy 100 clients with balanced device-type distribution in heterogeneous settings. For Fed MobiLLM and those centralized fine-tuning baselines, the number of training epochs is set to 20. To ensure fair comparison, all experiments share the same configurations unless specified: FP16 precision, batch size 8, learning rate 5e-4, maximum sequence length 256, and 60 Mbps in-lab Wi-Fi transmission speed. Additionally, LoRA and Fed MobiLLM employ rank-64 low-rank trainable modules by default, while FwdLLM uses 300 global perturbations per iteration.

### 5.3 BASELINES FOR PERFORMANCE COMPARISON

We compare Fed MobiLLM with three baseline approaches: i) **FedPETuning** (Zhang et al., 2023) (hereafter **FL**): Implements standard FedAvg aggregation with local PEFT on devices. ii) **Fed-Bert** (Tian et al., 2022) (hereafter **SFL**): Extends FL with split learning, retaining only first/last transformer layers on devices while offloading intermediate layers to the server. iii) **FwdLLM** (Xu et al., 2024): Follows FL paradigm but replaces backpropagation with on-device perturbation training.

Each baseline is evaluated with two representative PEFT methods: i) **LoRA** (Hu et al., 2022): Inserts trainable low-rank matrices into frozen backbone networks. ii) **BitFit** (Zaken et al., 2021): Fine-tunes exclusively bias terms while freezing other pre-trained weights.

## 6 EVALUATION RESULTS AND ANALYSIS

### 6.1 COMPARATIVE ANALYSIS WITH FEDERATED FT BASELINES

We conduct comprehensive experiments to validate Fed MobiLLM's advantages in on-device resource efficiency, training efficiency, and fine-tuning performance across homogeneous and heterogeneous device environments. (See Appendix D for details of result computation.)

**On-Device Resource Efficiency.** As detailed in Table 1, Fed MobiLLM demonstrates superior resource efficiency across metrics in terms of on-device memory footprint, computation cost, and

Table 1: Comparison across methods: i) Per-device resource efficiency (memory/computation/communication); ii) Fine-tuning accuracy (centralized training vs. federated training); and iii) Time-to-Accuracy (TTA) under homogeneous (TX2/Xavier/AGX Xavier clusters) and heterogeneous (Mixed) device configurations. Task: RoBERTa-Base@MRPC. *Note: Fed MobiLLM uses uniform backbone models across devices in mixed settings for fairness.*

| Methods | On-device memory (GB) | On-device comp. (TFLOPs) | On-device comm. (MB) | FT Performance (Acc.) | | TTA@86.5(Mins) | | | |
|---|---|---|---|---|---|---|---|---|---|
| | | | | Centralized | Federated | TX2 (Hom.) | Xavier (Hom.) | AGX (Hom.) | Mixed (Heter.) |
| FL-LoRA | 1.18 | 395.7 | 31.1 | 91.5 | 86.9 | 54.2 | 41.6 | 35.7 | 49.3 |
| FL-BitFit | 1.02 | 388.1 | 23.1 | 90.7 | 86.8 | 49.7 | 39.1 | 32.2 | 35.6 |
| SFL-LoRA | 0.54 | 58.3 | 5437.5 | 91.5 | 87.8 | 782.1 | 763.2 | 741.3 | 777.3 |
| SFL-BitFit | 0.49 | 56.5 | 5429.4 | 90.7 | 87.2 | 774.2 | 751.2 | 722.9 | 767.9 |
| FwdLLM-LoRA | 0.42 | 407.4 | 22.8 | 91.5 | 87.1 | 41.2 | 28.9 | 25.4 | 40.3 |
| FwdLLM-BitFit | **0.38** | 391.5 | 16.2 | 90.7 | 87.4 | 32.7 | 22.9 | 20.2 | 30.3 |
| **Fed MobiLLM** | **0.38** | **2.7** | **1.1** | 89.6 | **88.1** | **5.7** | **6.0** | **5.1** | **5.9** |

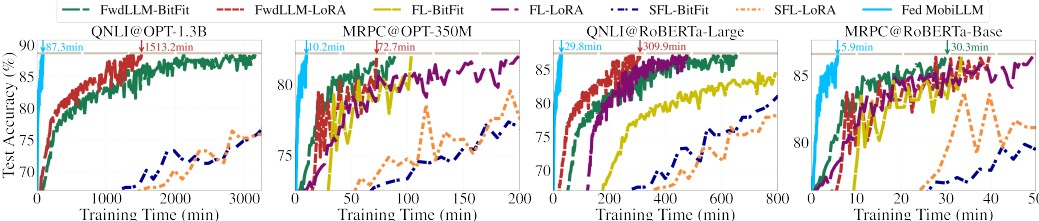

Figure 3: Convergence performance on various models and tasks under heterogeneous-device settings.

communication cost. Conventional FL-based PEFT methods (e.g., FedPETuning) incur substantial resource demands, while SFL approaches like FedBert trade computation savings for significantly increased communication overhead (up to $175\times$). Similarly, FwdLLM trades memory savings for higher computation load. In contrast, Fed MobiLLM maintains an optimal balance, where devices perform only a single forward pass, reducing memory consumption to inference levels (e.g., $2.68\times$ reduction for RoBERTa-Base). Besides, it achieves at least 95.2% lower computation and 93.2% less communication by eliminating on-device backpropagation and leveraging server-side activation caching.

**Training Efficiency.** We evaluate Fed MobiLLM's training efficiency by measuring time-to-accuracy across diverse configurations, including two model architectures (OPT, RoBERTa) and two tasks (MRPC, QNLI), as shown in Fig. 3. To ensure fair comparison, all heterogeneous devices use identical backbone models across Fed MobiLLM and other baselines. The results show that Fed MobiLLM achieves at least a $5.1\times$ speedup across all tasks. This acceleration stems from our full-pipeline efficient design: during the initial phase, clients perform only one forward propagation per data sample, avoiding both on-device backpropagation in conventional FL and multi-pass perturbations in FwdLLM. After aggregating activations from all devices, the server performs iterative training on the shared side-network independently, eliminating parameter synchronization with devices inherent in FL paradigms.

We further evaluate training efficiency across different client device setups, as shown in Table 1, which validates Fed MobiLLM's superior straggler resilience. Under the heterogeneous-device setting, baselines suffer severe slowdowns due to synchronous waiting periods in federated aggregation protocols, resulting in training times approaching the all-TX2 (lowest-capacity devices) configuration. In contrast, Fed MobiLLM benefits from extremely lightweight on-device computation and non-blocking parallel device-server collaboration. As a result, computational speed variations across devices are effectively masked, yielding consistent training times across diverse client setups.

**LLM FT Performance.** We evaluate Fed MobiLLM against federated LLM FT methods and their centralized PEFT counterparts (LoRA, BitFit, side-network tuning). As Table 1 shows, while LoRA and BitFit outperform side-network tuning in centralized settings, their accuracy significantly degrades under federated deployment with distributed data. Even the best federated baseline (FwdLLM-BitFit)

Table 2: FT Performance under data heterogeneity: IID vs. non-IID ($\alpha = \{0.1, 1.0, 10.0\}$).

| Methods | RoBERTa-Base@MRPC (Acc.) | | | | OPT-1.3B@DialogSum (R1/R2/RL) | | |
| | IID | non-IID ($\alpha$= 10.0) | non-IID ($\alpha$= 1.0) | non-IID ($\alpha$= 0.1) | IID | non-IID ($\alpha$= 10.0) | non-IID ($\alpha$= 0.1) |
|---|---|---|---|---|---|---|---|
| FL-LoRA | 86.9 | 85.1 | 84.5 | 82.1 | 19.2 / 6.1 / 15.1 | 17.9 / 5.2 / 14.3 | 16.1 / 4.6 / 12.7 |
| FL-BitFit | 86.8 | 84.9 | 84.6 | 81.5 | 19.0 / 6.2 / 14.9 | 17.7 / 5.1 / 14.2 | 16.4 / 4.2 / 12.9 |
| SFL-LoRA | 87.8 | 87.1 | 86.8 | 84.3 | 19.9 / 6.5 / 15.8 | 18.5 / 6.1 / 15.1 | 18.0 / 5.5 / 13.8 |
| SFL-BitFit | 87.2 | 87.0 | 86.1 | 85.2 | 19.7 / 6.2 / 15.5 | 18.7 / 6.0 / 15.2 | 17.8 / 5.4 / 13.4 |
| FwdLLM-LoRA | 87.1 | 86.7 | 86.5 | 84.1 | 19.3 / 6.3 / 15.7 | 18.2 / 6.2 / 15.1 | 17.8 / 5.2 / 13.7 |
| FwdLLM-BitFit | 87.4 | 87.1 | 86.6 | 84.7 | 19.6 / 5.9 / 15.1 | 18.6 / 6.1 / 14.9 | 17.3 / 5.1 / 14.1 |
| **Fed MobiLLM** | **88.1** | **87.8** | **87.7** | **87.3** | **21.0 / 7.7 / 17.2** | **20.2 / 7.4 / 16.0** | **20.0 / 7.3 / 16.3** |

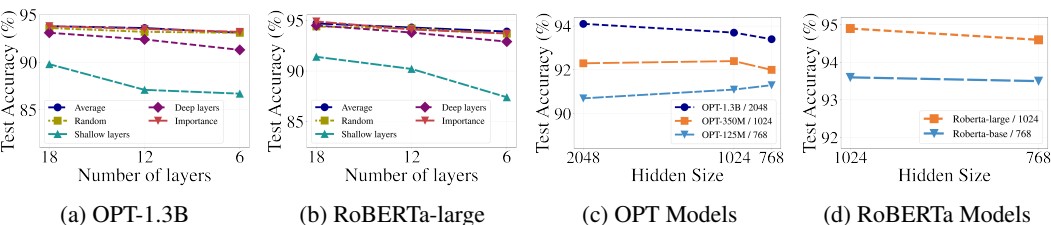

| (a) OPT-1.3B | (b) RoBERTa-large | (c) OPT Models | (d) RoBERTa Models |

Figure 4: Comparative accuracy across sampling methods (a-b) and hidden-size scaling (c-d) (Task: SST-2).

exhibits at least 3.3% performance drop. In contrast, Fed MobiLLM maintains near-centralized with only a 1.5% accuracy drop, outperforming all federated FT baselines by at least 0.3%.

We further evaluate the impacts of data heterogeneity on Fed MobiLLM's performance. To build local datasets, we use three Dirichlet distributions ($\alpha \in \{0.1, 1.0, 10.0\}$) where a lower $\alpha$ indicates a higher non-IID level (Zhang et al., 2023). Table 2 demonstrates that while the performance of all methods degrades under data heterogeneity, Fed MobiLLM shows superior resilience. Taking the MRPC task as an example, at extreme heterogeneity ($\alpha = 0.1$), Fed MobiLLM's accuracy drop is 1.2% smaller than SFL-BitFit, validating enhanced robustness to cross-client data distribution shifts.

## 6.2 CROSS-MODEL ALIGNMENT PERFORMANCE

We evaluate the efficacy of our layer-wise activation sampling and hidden size scaling methods for heterogeneous model adaptation through comparative experiments with alternative approaches.

**Layer Alignment Design.** To determine optimal block configurations for layer alignment in Fed MobiLLM, we perform comparative experiments using OPT-1.3B and RoBERTa-Large models. Specifically, we feed sampled layer-wise backbone activations to the side network and analyze how layer-wise activation sampling strategies affect performance, as shown in Fig. 4 (a-b). We compare five strategies: shallow-only, deep-only, average-interval, random, and importance-based selection (identified via layer-wise ablation). Results indicate: i) Performance improves with more backbone activation layers; ii) The average-interval strategy achieves accuracy comparable to computationally intensive importance-based approaches. These findings support Fed MobiLLM's configuration: set block count to the most lightweight LLM backbone's layer depth across devices, partitioning larger-sized LLMs at equal intervals to ensure structural alignment while maintaining performance.

**Hidden Size Alignment Design.** For cross-model hidden size scaling, determining a unified dimension for side-network adapters is critical. We experiment with various OPT and RoBERTa models, and evaluate performance under different side-network hidden sizes. As shown in Fig. 4 (c-d), peak performance is achieved when side-network hidden sizes match backbone sizes, while dimensional mismatches degrade accuracy. For example, OPT-125M (hidden size =768) performs worse when forced dimension to 2048, which demonstrates that larger dimensions aren't always beneficial. These findings yield practical Fed MobiLLM configuration guidelines: for multi-device scenarios, select median dimensions (e.g., 1024 for 768/1024/2048); for dual-model scenarios, prioritize larger dimensions to preserve representational capacity.

Table 3: Fed MobiLLM under heterogeneous device configurations with capacity-scaled backbone allocation. (*Single*: isolated side-network training on each single device's local data; *Global*: collaborative training of a shared side-network. Results partitioned by model family (OPT series upper, RoBERTa series lower) on SST-2 task.)

| Devices | Local Backbone Model | On-device Memory (GB) | On-device Comp. (TFLOPs) | Per-device Local Runtime(s) | FT Performance (Acc.) | | | | | |
|---|---|---|---|---|---|---|---|---|---|---|
| | | | | | IID | | ($\alpha$=1.0) | | ($\alpha$=0.1) | |
| | | | | | Single | Global | Single | Global | Single | Global |
| AGX | OPT-1.3B | 3.44 | 422.3 | 92.3 | 90.0 | 92.9 | 89.1 | 92.3 | 83.1 | 91.9 |
| Xavier | OPT-350M | 1.10 | 108.8 | 88.2 | 89.9 | 92.4 | 88.5 | 91.8 | 81.6 | 91.5 |
| TX2 | OPT-125M | 0.56 | 31.5 | 84.6 | 88.4 | 92.1 | 87.4 | 91.6 | 80.4 | 91.6 |
| AGX | RoBERTa-large | 0.86 | 108.4 | 81.9 | 91.8 | 93.5 | 88.1 | 92.7 | 84.3 | 91.7 |
| Xavier | RoBERTa-large | 0.86 | 108.4 | 92.4 | 91.8 | 93.5 | 88.1 | 92.7 | 84.3 | 91.7 |
| TX2 | RoBERTa-base | 0.38 | 30.9 | 90.7 | 90.4 | 92.7 | 87.6 | 92.3 | 82.9 | 91.0 |

## 6.3 VALIDATION OF HETEROGENEOUS BACKBONE ADAPTATION

To validate the efficacy and necessity of Fed MobiLLM's heterogeneous backbone design, we conduct systematic experiments across OPT and RoBERTa model series, where each device loads a capacity-scaled backbone model tailored to its hardware capabilities.

**Device-Specific Workload Balancing.** As shown in the results of the on-device workload in Table 3, device-specific backbone assignment optimizes hardware resource utilization compared to uniform model deployment. For example, AGX Xavier and Xavier run OPT-1.3B and OPT-350M, respectively, with memory usage both around 30% of their available capacities (12.4 GB and 4.6 GB). Computational loads similarly scale to device capacities, resulting in comparable execution times for local forward propagation and activation uploads across heterogeneous devices. This helps ensure near-balanced contributions to side-network training and prevent representation drift toward data from those fast devices. This confirms Fed MobiLLM's effective workload balancing and cross-client coordination through hardware-aware model scaling.

**Cross-Capacity Collaboration Effects.** We investigate whether collaboration with lower-capacity devices in Fed MobiLLM compromises high-capacity device performance. To this end, we evaluate performance under two settings: i) training the side network using only each device's local data (denoted by *Single*), and ii) federated training of a shared side network across all devices (denoted by *Global*). As shown in Table 3, *Global* consistently outperforms *Single* across different backbone sizes and data distributions, particularly under high data heterogeneity. For example, when $\alpha$ = 0.1, the accuracy improvement is at least 8.8% and 7.4% with OPT models and RoBERTa models, respectively. These results demonstrate that Fed MobiLLM enables all devices to benefit from collaboratively trained robust side-networks without performance degradation.

## 7 CONCLUSION

This paper has presented Fed MobiLLM, an efficient and scalable framework for federated fine-tuning of LLMs across heterogeneous mobile devices. By pioneering an asynchronous server-assisted side-tuning paradigm, Fed MobiLLM decouples device responsibilities to forward-only propagation and activation uploading, while the server trains a shared side-network, which eliminates synchronization bottlenecks inherent in conventional FL-based fine-tuning approaches. Through layer-wise activation sampling and cross-architecture dimension alignment, Fed MobiLLM enables each device to load backbone models that match its hardware capacities while still maintaining the ability to collaboratively train a shared side network, ensuring robust support for device heterogeneity. Extensive experiments demonstrate Fed MobiLLM's efficacy and efficiency: achieving $2.68\times$ reduction in on-device memory usage, 95.2% reduction in computational cost, 93.2% lower communication overhead, and $5.1\times$ faster convergence compared to state-of-the-art methods, while maintaining competitive accuracy under IID/non-IID data distributions. These results collectively establish Fed MobiLLM as a practical and deployment-ready solution for real-world federated LLM fine-tuning on distributed mobile datasets. (See Appendix E for extended discussion.)

# 8 REPRODUCIBILITY STATEMENT

It is important to note that the work presented in this paper is reproducible. To ensure the reproducibility of our results, we have made several efforts, which we summarize below. A detailed description of our method is provided in Section 4 and Appendix B. Comprehensive implementation details, encompassing hyperparameter configurations and optimization procedures, are delineated in Section 5 and Appendix D. For reproducibility, the source code has been included in the supplementary materials. Following acceptance, it will be released publicly on GitHub to facilitate further research. By providing these detailed resources, we aim to ensure that our work can be reproduced accurately. Furthermore, we encourage others to conduct further exploration and research based on our work.

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

## A  LLM USAGE DECLARATION

Large language models (e.g., ChatGPT) were used solely for language editing and formatting. They did not contribute to the conception, design, implementation, analysis, data generation or labeling, or evaluation of the methods and results. All technical content and claims were authored and verified by the authors, and no personal, proprietary, or sensitive data were shared with LLM services.

## B  FED MOBiLLM DETAILED PROCEDURES

Without loss of generality, we present here a more detailed description of the entire process by which Fed MobiLLM performs LLM fine-tuning for downstream tasks in heterogeneous mobile device scenarios.

*(1) Initialization (on mobile device & server):*

- Before federated LLM fine-tuning begins, each mobile device loads a pre-trained LLM backbone that matches its local compute and memory budget, chosen based on its inference footprint. Each device then sends the loaded backbone parameters to the server.

- Upon aggregating the configurations of all $N$ participating devices, the server configures the shared side network as follows: (i) set the number of adapter modules to the minimum backbone depth across devices (e.g. 12 blocks for 12/24-layer backbones); (ii) set the adapter hidden size to the median hidden dimension across devices (e.g., 1024 for 768/1024/2048); and (iii) allocate a projection layer $P$ for each backbone hidden size. The adapter modules and projection layers on the server are trainable and initialized from a zero-mean Gaussian distribution with a well-chosen standard deviation.

- The server then communicates the chosen number of shared adapters to all devices (i.e., the number of blocks to which each device will align). Given its own depth, each device determines which layers to upload by sampling in a block-wise manner at uniform intervals (e.g., every other layer in a 24-layer backbone).

*(2) Local backbone forward propagation and activation upload (on mobile device):*

- During training, the $N$ devices run independently. Each device samples a mini-batch from its local dataset and performs forward propagation through the frozen backbone. Following the block-wise selection decided at initialization, the device records activations at the transformer layers designated for upload.

- For each mini-batch, after the forward pass the device obtains the local prediction $y_{\text{pre}}$ and computes the deviation from the ground truth $y_{\text{label}}$ as $\Delta y = y_{\text{label}} - y_{\text{pre}}$. In parallel, for intermediate activations at each transformer layer, the device extracts only the positions relevant to the current task involved in the calculation of the loss (e.g., the last token in the classification), resulting in the activation set $(\mathbf{A}_i^1, \ldots, \mathbf{A}_i^L)$. For a more detailed explanation of $\Delta y$ and the token selector, refer to PAE MobiLLM (Yang et al., 2025). Once $\Delta y$ is computed and $(\mathbf{A}_i^1, \ldots, \mathbf{A}_i^L)$ is selected, the mini-batch forms an activation-deviation pair $(\mathbf{A}_i^1, \cdots, \mathbf{A}_i^L, \Delta \mathbf{y}_i)$, which is immediately submitted to the server.

- On each device, training proceeds mini-batch by mini-batch: after completing the forward pass and uploading the activation-deviation pair for a mini-batch, the device immediately moves to the next one. Once the local dataset has been traversed once, the device's local work is complete. In contrast to previous methods that repeatedly iterate over local data until convergence, Fed MobiLLM keeps the device-side backbone frozen and leverages server-side caching and reuse, so the device avoids redundant on-device computation.

*(3) Forward and backward propagation training (on server):*

- The server runs in an asynchronous, arrive-and-train manner: it receives activation–deviation pairs from devices and updates the model immediately upon arrival. For each incoming sample, the server first inspects the hidden size of the uploaded activations, applies the corresponding projection layer $p$ in the forward pass, and then runs the shared side-network

(adapters) forward. It computes the loss between the side-network output $y_{\text{side}}$ and the deviation $\Delta y$, and backpropagates along the same path. As a result, activations from different hidden-size backbones *jointly* train the shared adapters while *separately* updating their size-specific projection layers; activations with the same hidden size *jointly* update the same adapter stack and projection layer.

- After each update, the server also inserts the sample into a cache for replay. To mitigate non-IID bias, cached samples are randomly shuffled at insertion and sampling. During idle periods, the server continues to train on cached samples to maximize compute utilization. This design keeps training uninterrupted despite slow devices, eliminates straggler bottlenecks, and maintains high server utilization.

***(4) Fine-tuned side-network download and local inference (server $\rightarrow$ mobile device ):***

After the federated LLM fine-tuning is complete, the mobile device downloads from the server the **projection layer** p that matches its hidden size and the **side-network (adapters)**, for on-device inference (see Fig. 5). These modules can be seamlessly attached to the frozen local backbone. During inference, the side network produces $y_{\text{side}}$, which

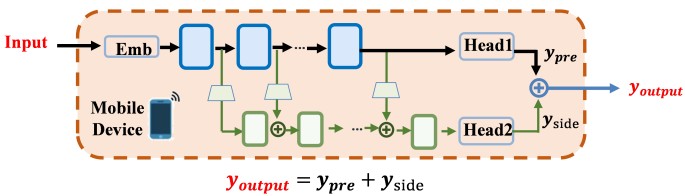

Figure 5: How to execute on-device inferences in Fed MobiLLM.

provides a residual correction to the backbone output $y_{\text{pre}}$, producing the fine-tuned model output $y_{\text{output}} = y_{\text{pre}} + y_{\text{side}}$. This also clarifies why the server-side training targets the deviation $\Delta y = y_{\text{label}} - y_{\text{pre}}$: by learning to predict $\Delta y$, the deployed side-network output $y_{\text{side}}$ approximates this deviation, ensuring the residual correction is aligned with the downstream task.

## C  DATASETS STATISTICS

Table 4: Datasets Statistics.

| Dataset | Description | Task | # Samples (train/eval) |
|---|---|---|---|
| CoLA | Linguistic Acceptability | Classification | 8551 / 1043 |
| SST2 | Sentiment Analysis | Classification | 67350 / 873 |
| MRPC | Sentence Equivalence | Classification | 5801 / 408 |
| STSB | Sentence Similarity | Regression | 5712 / 1471 |
| QQP | Paraphrase Recognition | Classification | 363847 / 40431 |
| RTE | Textual Entailment | Classification | 2491 / 278 |
| QNLI | Natural Language Inference | Classification | 103141 / 5268 |
| MNLI | Textual Entailment | Classification | 392702 / 9815(9832) |
| DialogSum | Abstract Summary | Generation | 12460 / 500 |

## D  COMPUTATION OF REPORTED RESULTS

In this section, we use Table 1 as an example to present how we compute *on-device resource overhead* and *training efficiency* metrics for Fed MobiLLM and other baselines. We consider a federated setting with 100 homogeneous NVIDIA Xavier clients, each of which loads the same RoBERTa-Base backbone. The unified experimental setup is as follows: using the MRPC dataset with 5,801 samples, data is evenly distributed across 100 clients (58 samples per client) according to the standard federated learning configuration. FP16 precision, batch size = 8, sequence length = 256, and an in-lab Wi-Fi throughput of 60 Mbps.

### D.1  ON-DEVICE MEMORY FOOTPRINT

- **FL-LoRA:** Performs full LoRA fine-tuning on the RoBERTa model on the device. The memory footprint consists of model weights + intermediate activation + optimizer states and hence is the largest among all methods.

- **SFL-LoRA (U-shaped split learning):** To keep raw data and labels on the device, only offload the middle 10 layers of the 12-layer transformer to the server; the remaining layers are LoRA fine-tuned on the device. The memory footprint consists of partial weights + partial intermediate activation + partial optimizer states, thus lower than FL-LoRA.

- **FwdLLM-LoRA and Fed MobiLLM:** The device only performs forward propagation. The memory footprint consists of model weights + a small inference-time intermediate activation, achieving the smallest memory footprint.

### D.2  ON-DEVICE COMPUTATION

We report on-device computation as the total FLOPs executed on a single device over the entire training process until it reaches the target accuracy. Let the number of global rounds be $R$, the number of local iterations (epochs/passes) per round be $E$, and let $\mathrm{Cost}_{\mathrm{local}}$ denote the computation for one complete local epoch under a given method.

- **FL-LoRA:** Each local epoch performs the full RoBERTa + LoRA (forward & backward).
$$\mathrm{Computation}_{\mathrm{device}} \;=\; \mathrm{Cost}_{\mathrm{FT(full\ backbone,\ LoRA)}} \times R \times E.$$

- **SFL-LoRA (U-shaped split learning):** Only the non-offloaded layers performs LoRA on device (forward & backward).
$$\mathrm{Computation}_{\mathrm{device}} \;=\; \mathrm{Cost}_{\mathrm{FT(partial\ backbone,\ LoRA)}} \times R \times E.$$

- **FwdLLM-LoRA (forward-only with perturbations):** The device performs only forward propagation; each round uses $K$ local perturbation forwards.
$$\mathrm{Computation}_{\mathrm{device}} \;=\; \mathrm{Cost}_{\mathrm{Infer(full\ backbone)}} \times R \times K.$$

- **Fed MobiLLM:** The device performs a single forward propagation traversal of its local data (no local replay).
$$\mathrm{Computation}_{\mathrm{device}} \;=\; \mathrm{Cost}_{\mathrm{Infer(full\ backbone)}} \cdot$$

### D.3  ON-DEVICE COMMUNICATION

We report on-device communication as the total amount of data a single device exchanges with the server over the entire training process until it reaches the target accuracy. Let the number of global rounds be $R$, the number of local iterations (epochs/passes) per round be $E$.

- **FL-LoRA:** In standard FL, devices and server exchange trainable parameters in both directions (upload/downlink) each global round:
$$\mathrm{Communication}_{\mathrm{device}} = 2 \times R \times \left|\theta_{\mathrm{LoRA}}\right|,$$
where $\left|\theta_{\mathrm{LoRA}}\right|$ is the size of the on-device LoRA trainable parameters.

- **SFL-LoRA (U-shaped split learning):** Beyond the round-wise parameter exchange, split learning requires frequent exchange of intermediate activations and backward gradients during local forward/backward:
$$\mathrm{Communication}_{\mathrm{device}} = 2 \times R \times \left|\theta_{\mathrm{LoRA}}\right| \;+\; R \times E \times \mathrm{Comm}_{\mathrm{act/grad\text{-}per\ epoch}},$$
where $\mathrm{Comm}_{\mathrm{act/grad\text{-}per\ epoch}}$ denotes the activation/gradient traffic for one local forward+backward epoch.

- **FwdLLM-LoRA.** Similar to FL-LoRA, the standard communication rhythm for federated learning:
$$\mathrm{Communication}_{\mathrm{device}} = 2 \times R \times \left|\theta_{\mathrm{LoRA}}\right|.$$

- **Fed MobiLLM:** The device performs a single forward propagation traversal of its local data and uploads only a small subset of layer-wise activations selected by the token selector; there is no parameter round-trip or gradient return:
$$\mathrm{Communication}_{\mathrm{device}} \;\approx\; \mathrm{Comm}_{\mathrm{selected\text{-}activations\text{-}\ single\ epoch}} \cdot$$

### D.4 Training Time to Target Accuracy

We report the training time to target accuracy for each LLM federated FT framework. Let the number of global rounds be $R$, and define: $t_1$ — per-round on-device runtime, $t_2$ — per-round device–server parameter communication time, $t_3$ — per-round server aggregation time.

For **FL-LoRA / SFL-LoRA / FwdLLM-LoRA**, the per-round steps run *serially*, so

$$T_{\text{train}} = (t_1 + t_2 + t_3) \times R.$$

The only difference lies in $t_1$:

- **FL-LoRA:** $t_1$ = on-device LoRA fine-tuning (forward + backward).
- **SFL-LoRA:** $t_1$ = split-learning based device–server co-training; frequent activation/gradient exchange makes $t_1$ typically longer.
- **FwdLLM-LoRA:** $t_1$ = on device multiple forward propagations.

**Fed MobiLLM** does not follow a federated sequential rhythm. Instead, it overlaps single forward propagation time on the device side with communication time and server iteration time in parallel. The total training time approximates the maximum duration among these three components. In our testing, the server iteration typically dominates, i.e., $T_{\text{train}} \approx t_{\text{server-iter}}$.

## E Discussions

### E.1 Impact of Network Speed on Training Efficiency

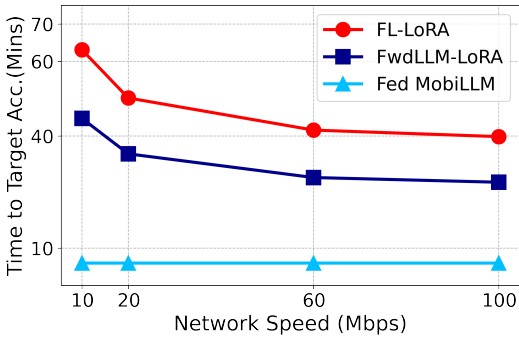

Figure 6: Impact of network speed on time-to-accuracy (homogeneous Xavier clusters, task: RoBERTa-Base@MRPC).

In practical federated LLM fine-tuning, device-server communication makes training efficiency sensitive to network conditions. Consequently, network fluctuations may impact overall training efficiency. Figure 6 illustrates the training efficiency across methods under different wireless transmission speeds. Since the total training time in FL-LoRA and FwdLLM-LoRA is affected by the communication time, they suffer severe performance degradation under low-speed transmission conditions. For example, FwdLLM-LoRA shows a $1.62\times$ slowdown at 10Mbps vs. 100Mbps. In contrast, Fed MobiLLM maintains stable total latency across different transmission speeds, demonstrating almost no variations from 10 Mbps to 100 Mbps. Such resilience stems from parallel scheduling that overlaps communication with device- and server-side computation, masking transmission delays with on-device compute time that is unaffected by network-speed fluctuations.

### E.2 Server Storage

As shown in Table 5, we report Fed MobiLLM's server-side cache usage across different backbone sizes (from RoBERTa-Base to OPT-1.3B) and dataset scales (from RTE to QNLI). When targeting larger backbones or higher data throughput, edge servers with tight storage budgets may require

Table 5: Server cache size across backbones and tasks.

| Backbone | Server cache size (GB) | | |
| --- | --- | --- | --- |
| | RTE | SST2 | QNLI |
| OPT-1.3B | 0.23 | 6.21 | 36.21 |
| RoBERTa-Base | 0.08 | 2.40 | 13.99 |

additional scalability measures for the caching mechanism. As future work, we will explore activation-aware quantization (e.g., mixed 2–4-bit precision per layer) and intelligent lifecycle management (automatically purging stale or low-impact activation–deviation samples) to align storage costs with operational budgets without compromising adaptation quality.

