# OpenReview forum: "Fed MobiLLM: Efficient Federated LLM Fine-Tuning over Heterogeneous Mobile Devices via Server Assisted Side-Tuning"
_ICLR.cc/2026/Conference — ICLR 2026 Conference Withdrawn Submission_

### Official Review · Reviewer_nC9y · 2025-10-14

**Soundness:** 3
**Presentation:** 3
**Contribution:** 1
**Rating:** 2
**Confidence:** 4

**Summary:**

Fed MobiLLM proposes a hybrid distributed training framework designed for federated settings involving heterogeneous devices. In this approach, client devices are responsible only for the forward pass over their local data, while the server performs the computationally expensive backward pass and gradient updates using the activations transmitted from clients. This design aims to reduce the client-side computational and memory burden while retaining data locality. The work further investigates this paradigm under both device and data heterogeneity conditions.

**Strengths:**

1.	Comprehensive Evaluation under Heterogeneity: The paper evaluates Fed MobiLLM under both device heterogeneity (e.g., varying model capacities) and data heterogeneity (non-IID distributions), demonstrating attention to realistic federated scenarios that many existing works overlook.

2.	Relevance: The topic of reducing on-device resource consumption in federated learning is highly relevant, especially given the growing interest in deploying large models on edge or mobile devices.

3.	Clarity of Presentation: The paper is well written and clearly explains the technical setup and experimental design, making the methodology easy to follow.

**Weaknesses:**

1.	Weak Motivation and Questionable Privacy Rationale: The central motivation (delegating only the backward pass to the server) needs stronger justification. If privacy of the data is the primary concern, transmitting activations and labels to the server already exposes rich representational information that can be inverted to approximate original data (as shown in prior work on gradient and activation leakage attacks [1, 2]). Thus, the privacy gain from not sending raw data is limited. The authors should clarify the privacy–utility trade-off and potentially quantify the degree of privacy leakage through experiments or formal guarantees.

Other than that, since running inference/a forward pass is not computationally expensive anyway, the motivation behind using the federated setting and using clients only for inference is weak.

2.	Insufficient and Non-competitive Baselines: The baseline comparison overlooks recent strong alternatives such as MeZO [3]/FedMeZO [4], which employ zero-order gradient estimation to enable fully local training without gradient computation, and Spry [5], which uses forward-mode automatic differentiation to achieve favorable accuracy–efficiency trade-offs. Both represent valid, state-of-the-art strategies for reducing on-device computation, yet they are not discussed or compared. Consequently, the claimed advantages of Fed MobiLLM are not convincingly demonstrated.

3.	Lack of Conceptual Novelty: The idea of server-side backpropagation using client-side activations is not new; similar frameworks (e.g., SplitFed [6], Split Learning, and other hybrid FL paradigms [7]) already transfer intermediate activations to the server to handle backward computation. The only novelty appears to be the adaptation of the MobiLLM paradigm (previously developed for centralized settings) into a federated context. Without additional algorithmic or system-level contributions, the work's originality is limited.

4.	Missing Comparison with the Centralized Counterpart (PAE MobiLLM): Since Fed MobiLLM is derived from PAE MobiLLM, a direct quantitative and qualitative comparison with its centralized variant would be valuable to isolate the effects of federation. This would clarify whether performance degradation arises from the federated setup, communication constraints, or model partitioning.

5.	Misaligned Baselines for Data Heterogeneity: The baselines in Table 2 appear to focus on performance or memory/compute efficiency rather than robustness to non-IID data, making the comparison partially inappropriate. Moreover, the proposed method does not include any specific mechanisms (e.g., personalization layers, regularization, or feature alignment) to directly mitigate data heterogeneity—merely sampling activations from multiple clients does not constitute a principled treatment of the problem.

[1] Zhu, L., Liu, Z., & Han, S. (2019). Deep Leakage from Gradients. NeurIPS.

[2] Yin, H., et al. (2021). See through Gradients: Image Batch Recovery via GradInversion. CVPR.

[3] Malladi, S., et al. (2023). Fine-Tuning Language Models with Just Forward Passes. NeurIPS.

[4] Ling, Z., et al. (2024). On the convergence of zeroth-order federated tuning for large language models. SIGKDD .

[5] Panchal, K., et al. (2024). Thinking forward: memory-efficient federated finetuning of language models. NeurIPS.

[6] Thapa, C., et al. (2022). SplitFed: When Federated Learning Meets Split Learning. AAAI.

[7] Kairouz, P., et al. (2021). Advances and Open Problems in Federated Learning. Foundations and Trends in Machine Learning. FTML.

**Questions:**

1.	The paper states that "to mitigate non-IID data bias, cached samples are randomly shuffled during storage." It is unclear how this mechanism effectively addresses data heterogeneity. Random shuffling of activations at the server might decorrelate sample ordering, but it does not alter the underlying distributional imbalance among clients. A more robust solution would require either feature-level alignment, domain-invariant representations, or weighted aggregation strategies to counter non-IID effects.

2.	It would be helpful to clarify whether each client performs the forward pass for a single epoch per communication round or multiple epochs (e.g., 20). Furthermore, can this number adapt dynamically to client compute capabilities? A flexible scheduling policy could improve overall system efficiency but might introduce gradient staleness or imbalance across clients. The paper should discuss how such trade-offs are managed.

---

### Official Review · Reviewer_Qbth · 2025-10-29

**Soundness:** 1
**Presentation:** 3
**Contribution:** 2
**Rating:** 2
**Confidence:** 4

**Summary:**

This paper focuses on the efficiency of federated LLM fine-tuning among mobile devices. The authors aim to address the memory limitation and computation bottleneck faced by mobile devices. They propose to freeze the model on the devices and only update the global model on the central server. During training, the devices only perform forward propagation and send the activations of each layer to the server for model updates. After training, the authors proposes a heuristica method to reconstruct the client models.

**Strengths:**

The problem studied in this work is very interesting. Federated fine-tuning of LLM among mobile devices faces the memory and computation challenges. Addressing these issues is definitely important.

This work is well written and the system diagram looks cool, which helps me to understand this work.

**Weaknesses:**

1. The proposed methodology seems to contradict the fundamental motivation of federated learning. The key idea in this work is to transmit intermediate activations from clients to the server, which then performs all parameter updates. Conceptually, this is nearly equivalent to uploading the raw data to a central server and performing centralized training. In the homogeneous model setting, following the authors’ logic, an even simpler and more efficient alternative would be to send the activations of the first layer together with the labels; from these, the server could in principle reconstruct all subsequent intermediate representations on its own.


2. The solution for handling model heterogeneity appears entirely heuristic. Moreover, it introduces a substantial number of additional parameters on the server side through the per-client projection layers, whose total size scales linearly with the number of clients. A simple estimation illustrates the problem: with 100 clients and 12 layers, assuming each projection matrix is of size $1080 \times 1080$, the total number of parameters reaches approximately $1.4 \times 10^9$, which actually exceeds the number of parameters in the base model itself. In the context of LLM fine-tuning with limited client data, it is unclear how such a large number of projection parameters can be trained effectively from scratch. Even though some projection may be shared across clients, but the training of introduced projection matrices are not well justified.

3. The model at the client side is frozen, and there is no information transmission from the server to the clients. Therefore, the model updates on the server will have no impact on the local side. This seems strange in the context of federated learning. In other words, the clients only need to process their data once; afterward, they do not need to perform any further computation since the model is frozen and the output for the same input would be time-invariant. I am skeptical about this design.


4. The paper argues that existing federated fine-tuning methods suffer from computation and memory bottlenecks on client devices. However, gradient checkpointing and parameter-efficient techniques such as LoRA largely mitigate these issues in practice. In contrast, the proposed algorithm requires each client to store and transmit intermediate activations of multiple layers for each token. This implies that the memory footprint on clients may in fact be larger than that of baseline methods, which can flexibly apply checkpointing and gradient recomputation. Thus, the claimed efficiency advantage of FedMobiLLM is not convincing to me.

**Questions:**

In the experiments, the authors mention that they adopted 100 NVIDIA Jetson devices (Line 298) as mobile clients and an A100 GPU as the server. I have a few concerns and questions about this setup.

Did you actually deploy 100 physical Jetson devices for the simulation? If so, how were these devices networked and how did they communicate with the A100 server (e.g., through local Ethernet or Wi-Fi)?

If this was only a simulated setup, why not directly conduct the entire experiment on a GPU cluster? Since the forward outputs of each layer on Jetson and A100 should be nearly identical (aside from minor numerical differences), the use of physical Jetsons seems unnecessary. Do you want to model the unstable communciation or device outage?

---

### Official Review · Reviewer_adVV · 2025-10-30

**Soundness:** 2
**Presentation:** 2
**Contribution:** 2
**Rating:** 2
**Confidence:** 4

**Summary:**

This paper proposes FedMobiLLM to more efficiently finetune LLMs across the heterogeneous mobile devices with different system capacities. The paper proposes to save backbone of the LLM frozen on the device and only communicate the intermediate activations for training. The devices only perform forward propagation on their local data using only this frozen backbone LLM, and once done with the forward activation, sends the updates to the server that updates its model asynchronously using the activations to perform the backpropagation. FedMobiLLM also allows the mobile devices to have different model architectures frozen on their deice to take into consideration of heterogeneity amongst the system capacities.

**Strengths:**

- This work investigates how to train LLMs on federated learning more efficiently which is indeed a relevant topic to look into since devices are bottlenecked by their system capacity.
- The work is written clearly and reads well.
- The work has evaluated FedMobiLLM in many different aspects including reduction in on-device memory usage, computation cost, communication overhead, and convergence with different heterogeneity settings.
- The work also particularly gives care on the reproducibility of the work by giving a more detailed explanation of the work with ablation studies in the appendix and open sourced the code on github.

**Weaknesses:**

- The paper is unclear on how the asynchronous updates actually differ from the standard vanilla FedAvg algorithm mathematically, and its guarantee of convergence. As the exact update rule for each client & server is not really laid out in the paper I'm unclear how this can affect the standard convergence of the algorithm of FedAvg. For instance, what if the global model that is frozen on the clients' device is stale, but the global model keeps updating the model with the local updates it receives?
- Freezing the backbone of the LLM and deferring the calculation to the server, as the authors also point out, has already been proposed in previous work such as in PAE MobiLLM (Yang et al., 2025). The main novelty seems to be the asynchronous updates on the server, which I have doubts about as mentioned above.
- I'm also concerned of how the data heterogeneity and different model deployment across the devices may interact with each other, and potentially make the global model more biased towards the clients that have larger models, as observed in previous work such as in [1]. In Table 2, it is weird to be that there is not much difference across the performance of the clients depending on the degree of data heterogeneity.


[1] Heterogeneous Low-Rank Approximation for Federated Fine-tuning of On-Device Foundation Models, YJ Cho, L Liu, Z Xu, A Fahrezi, G Joshi, EMNLP 2024

**Questions:**

Please address the weaknesses above.

---

### Official Review · Reviewer_usFr · 2025-10-31

**Soundness:** 2
**Presentation:** 3
**Contribution:** 2
**Rating:** 4
**Confidence:** 4

**Summary:**

This paper proposes Fed MobiLLM, a server-assisted federated side-tuning framework to enable LLM fine-tuning on resource-constrained mobile devices. The method offloads backpropagation to the edge server while devices perform only forward inference, reducing computation, memory, and synchronization costs. A layer-wise activation alignment mechanism addresses model heterogeneity. Experiments on commercial edge-device testbeds show significant reductions in memory, communication, and convergence time while maintaining accuracy.

**Strengths:**

1. This paper targets an important and timely problem. Enabling efficient federated fine-tuning of LLMs on resource-constrained edge devices is a critical and emerging challenge for distributed AI systems.

2. The proposed server-assisted side-tuning framework is conceptually novel. By decoupling client-side forward computation from server-side backpropagation, the design effectively addresses both device heterogeneity (computation/memory) and model heterogeneity (different backbone architectures).

3. The experimental evaluation covers training efficiency, memory footprint, communication overhead, and heterogeneity robustness. Results demonstrate consistent and clear improvements over existing baselines.

4. This paper combines algorithmic and system-level optimizations, introducing mechanisms such as layer-wise alignment and hidden-size scaling. Moreover, the system is implemented and evaluated on commercial edge-device testbeds, providing convincing evidence of real-world feasibility.

**Weaknesses:**

1. The motivation is present but not sufficiently compelling due to a lack of detailed descriptions and quantitative evidence. For instance, while the authors mention the memory overhead of large-scale models (line 50–52), they do not specify whether quantization techniques or bit-width configurations were used, which are crucial in real-world edge deployment. In addition, the motivation (Section 3) mainly discusses theoretical limitations of existing methods but lacks quantitative or visual comparisons to highlight the practical performance gaps.

2. The proposed Fed MobiLLM lacks sufficient theoretical analysis or deeper conceptual insight. Although it performs well empirically, the overall design leans more on heuristic methods and system-level optimization than a principled algorithmic study, making the approach somewhat experimental in nature. Providing more analytical understanding of the underlying mechanisms would further strengthen the contribution.

3. This paper lacks sufficient discussion of several key related works. Recent advances in heterogeneous-rank adaptation methods such as FLoRA [1] and FlexLoRA [2], as well as system-level approaches like LEGEND [3], are not adequately compared or discussed. Moreover, considering the increasing relevance of quantization-based optimization (e.g., FAH-QLoRA [4], HAFLQ [5]) and dynamic parameter pruning methods such as FedMef [6], the paper would benefit from a more comprehensive positioning within the broader landscape of federated and resource-efficient LLM fine-tuning.

[1] Wang et al. FLoRA: Federated Fine-Tuning Large Language Models with Heterogeneous Low-Rank Adaptations. NuerIPS’2024.
[2] Bai et al. Federated Fine-tuning of Large Language Models under Heterogeneous Tasks and Client Resources. NuerIPS’2024.
[3] Liu et al. Adaptive Parameter-Efficient Federated Fine-Tuning on Heterogeneous Devices. IEEE TMC’2025.
[4] Gao et al. Federated Adaptive Fine-Tuning of Large Language Models with Heterogeneous Quantization and LoRA. INFOCOM’2025.
[5] Su et al. Heterogeneous Adaptive Federated LoRA Fine-tuned LLM with Quantization. ArXiv’2024.
[6] Huang et al. FedMef: Towards Memory-efficient Federated Dynamic Pruning. CVPR’2024.

4. This paper aims to address federated fine-tuning for future large-scale LLMs, yet the experiments rely mainly on smaller and relatively outdated models (RoBERTa and OPT), which weakens the overall persuasiveness. Moreover, the evaluation is limited to language understanding and summarization tasks, lacking broader benchmarks such as MMLU, GSM8K, and HumanEval that better reflect general reasoning and frontier capabilities.

**Questions:**

1. Compared with conventional methods that upload gradients or model parameters, does transmitting intermediate activations pose a higher risk of privacy leakage? If so, how does the proposed framework mitigate this potential issue?
2. The communication bottleneck might be underestimated, since the size of uploaded activations can increase rapidly with deeper hidden layers or larger batch sizes. How would the proposed method handle this scalability issue in practice?
3. Since all backpropagation is performed on the server, the computational load may shift heavily to the central server. How is multi-client parallel backpropagation managed efficiently in this setup?
4. When the number of participating devices increases, server-side computation or GPU memory may become a bottleneck. Additionally, does the one-step asynchronous update scheme affect the global convergence consistency?

---

### Note · Authors · 2025-11-18

I have read and agree with the venue's withdrawal policy on behalf of myself and my co-authors.